# Structural basis of lysophosphatidylserine receptor GPR174 ligand recognition and activation

Jiale Liang[1,4], Asuka Inoue [2,4] ✉, Tatsuya Ikuta [2], Ruixue Xia[1], Na Wang[1], Kouki Kawakami [2], Zhenmei Xu[1], Yu Qian[1], Xinyan Zhu[1], Anqi Zhang[3], Changyou Guo[3], Zhiwei Huang [3] & Yuanzheng He [1] ✉

Lysophosphatidylserine (LysoPS) is a lipid mediator that induces multiple cellular responses through binding to GPR174. Here, we present the cryo-electron microscopy (cryo-EM) structure of LysoPS-bound human GPR174 in complex with $G_s$ protein. The structure reveals a ligand recognition mode, including the negatively charged head group of LysoPS forms extensive polar interactions with surrounding key residues of the ligand binding pocket, and the L-serine moiety buries deeply into a positive charged cavity in the pocket. In addition, the structure unveils a partially open pocket on transmembrane domain helix (TM) 4 and 5 for a lateral entry of ligand. Finally, the structure reveals a $G_s$ engaging mode featured by a deep insertion of a helix 5 (αH5) and extensive polar interactions between receptor and αH5. Taken together, the information revealed by our structural study provides a framework for understanding LysoPS signaling and a rational basis for designing LysoPS receptor-targeting drugs.

Lysophosphatidylserine (LysoPS) is a hydrolyzed product from phosphatidylserine (PS) via phospholipase $A_1$ and $A_2$ at either sn-1 or sn-2 position[1,2]. LysoPS has been shown to act like lipid mediators to regulate a broad of physiologies, including mast cell degranulation, neurite growth, fibroblast migration, and suppression of T lymphocyte proliferation[3–6]. LysoPS exerts its physiological roles through binding and activating LysoPS receptors, members of the G-protein-coupled receptor (GPCR) superfamily. The first identified LysoPS receptor was GPR34 (also named as $LPS_1$), a class A GPCR capable of $G_i$ coupling[7,8]. GPR34 is highly expressed in microglia and plays a protective role against external pathogen infection in the central nervous system. Later, P2Y10 and GPR174 were identified as second and third LysoPS receptors (also named $LPS_2$ and $LPS_3$, respectively), through an in vitro screen via the transforming growth factor-α (TGF-α) shedding assay[9,10]. P2Y10 couples almost exclusively to $G_{12/13}$ pathway while GPR174 couples both $G_{12/13}$ and $G_s$ signaling[10,11]. Both P2Y10 and GPR174 are predominantly expressed in lymphoid organs such as the thymus, spleen, and lymph nodes, suggesting the immunological roles of the two receptors. In support of this, GPR174 was found to be involved in suppression of interleukin-2 production and CD4 T cells[12]. Intriguingly, GPR174 was reported to be abundantly expressed in developing and mature regulatory T cells ($T_{reg}$) and negatively regulates $T_{reg}$ cell accumulation[13]. These T-cell suppressive effects were shown to be mediated by $G_s$ signaling. Since $T_{reg}$ has been indicated in tuning down the excessive and overwhelming inflammatory response, antagonism of GPR174 may have therapeutic potential for autoimmune disease. Furthermore, studies found that nonsynonymous single nucleotide polymorphisms (SNPs) of GPR174 are risk factors of autoimmunity, including Graves's disease and Addison's disease[14,15].

LysoPS is a member of lysophospholipids (LysoGPs), deacylated forms of phospholipids with a single fatty acid chain. Interestingly, many LysoGPs serve as signaling molecules, including

[1]Laboratory of Receptor Structure and Signaling, HIT Center for Life Sciences, School of Life Science and Technology, Harbin Institute of Technology, 150001 Harbin, China. [2]Graduate School of Pharmaceutical Sciences, Tohoku University, 6-3, Aoba, Aramaki, Aoba-ku, Sendai 980-8578 Miyagi, Japan. [3]HIT Center for Life Sciences, School of Life Science and Technology, Harbin Institute of Technology, Harbin, China. [4]These authors contributed equally: Jiale Liang, Asuka Inoue. ✉e-mail: iaska@tohoku.ac.jp; ajian.he@hit.edu.cn

lysophosphatidic acid (LPA), lysophosphatidylinositol (LPI), and sphingosine-1-phosphate (S1P)[1], to regulate a diversity of physiologies mostly through binding to their GPCRs. The structures of LPA-bound LPAR1 and S1P-bound S1PR1 have been recently reported[16,17], however, there is no LysoPS-bound receptor structure available, which hindered the understanding of LysoPS signaling and the development of LysoPS intervention for treatments of associated autoimmune diseases. For this reason, we solved the cryo-EM structure of LysoPS-bound human GPR174 in a complex with $G_s$. The structure unveils a ligand recognition model that is largely different from the reported lysophospholipids receptors. The structure also reveals a distinctive $G_s$-engaging model that has not been observed in other $G_s$-coupled receptors.

## Results

### The overall architecture of GPR174/$G_s$ complex

We adopted a NanoBiT tethering strategy[18] in the GPR174/$G_s$ complex assembling. To this end, we fused the C-terminus of GPR174 and the C-terminus of $G\beta_1$ with the large fragment and the high-affinity small fragment of NanoBiT, respectively (detail see the "Methods" section). For the $G_s$ protein, we use mini-$G\alpha_s$ adopted from the reported melanocortin receptor 1 (MC1R)/$G_s$ complex[19]. Together with a $G\gamma_2$ construct, we express all of the four components (GPR174-LgBiT, mini-$G\alpha_s$, $G\beta_1$-HiBiT and $G\gamma_2$) of the GPR174/$G_s$ complex in Sf9 insect cells and purify it via a conventional membrane protein purification method (Supplementary Fig. 1 and detail see the "Methods" section). LysoPS was added in the cell-lysing step and all of the subsequent purification procedures. Nb35, a $G_s$-stabilizing nanobody, was added to the purification process to increase the stability of the complex. We used the single particle analysis of cryo-EM to solve the complex structure at a resolution of 2.76 Å (Supplementary Figs. 2, 3 and Supplementary Table 1). Local resolution analysis shows that the receptor/$G_s$ interface, $G\beta_1$ core, and the interface of $G\alpha_s$, $G\beta_1$, and Nb35 have the highest electron density, while the extracellular side of receptor and helix 8 (H8) have relatively weak density. The map of the receptor is of high quality which allows us to resolve almost the entire receptor including the intracellular loop 3 (ICL3), except the first 14 residues of the N-terminus and residues after L303 of H8. The overall architecture of the GPR174/$G_s$ complex resembles the canonical GPCR/G-protein complex in which $G\alpha$ uses its C-terminus, mainly $\alpha$ helix 5 ($\alpha$H5) to engage the intracellular part of the receptor (Fig. 1).

### LysoPS recognition in GPR174

The density of the ligand is of high quality (Fig. 2a, b) which allows us to unambiguously assign the LysoPS (18:0) molecule into the ligand pocket. The ligand pocket is formed by the extracellular half of transmembrane helix 1–7 (TM1–7) and extracellular loop 2 (ECL2) (Fig. 2a and Supplementary Fig. 4a). We noticed that GPR174 has a long ECL2 which covers the entire extracellular opening of the ligand pocket (Supplementary Fig. 4a) and provides an extra binding site for the ligand (Fig. 2b). The acyl chain of LysoPS extends out of a cavity formed by the middle ridge of the extracellular side of TM4,5 and snuggles the rest of tail into a groove formed by the lower part of TM3,4,5 (Supplementary Fig. 4b). The partially open pocket may provide a lateral entry of ligand through the membrane side as seen in LPAR6[20]. Like other phospholipids, LysoPS is a zwitterionic molecule consisting of a charged polar head and a highly hydrophobic acyl tail, which matches well with the hydrophobicity of the ligand binding pocket where the front part of the pocket is hydrophilic and the middle and rear part of the pocket is highly hydrophobic (Supplementary Fig. 4c). Similarly, an electrostatic potential analysis shows that the negatively charged head group of LysoPS nicely fits into the positively charged front pocket, particularly, the L-serine edge of LysoPS head buries deeply into a highly positively charged cavity. On the other hand, the other part of the pocket is neutral and fits well with the acyl chain of LysoPS (Supplementary Fig. 4d).

The head group of LysoPS consists of L-serine, phosphodiester, glycerol, and ester linkage (Fig. 1 middle panel and Supplementary Fig. 9), they all make excellent polar interactions with surrounding polar residues of the ligand binding pocket (Fig. 2b, c). Specifically, the serine head interacts with Y99[3.33], R75[2.60], Y79[2.64], and the backbone carbonyl group of F169[ECL2]; the phosphate group forms salt bridges with K257[6.62] and R156[4.64]; the $sn$−2 hydroxyl group of the glycerol interacts with Y246[6.51]; the carbonyl group of the ester linkage forms polar interaction with R156[4.64] (Figs. 2b, c and 3b). Interestingly, we identified a water molecule in the upper pocket which connects R18[1.31] to the phosphate of LysoPS. In addition, Y22[1.35] and K98[3.32] are also in proximity of the charged head of LysoPS (Fig. 2b). We also observed hydrophobic interactions between LysoPS and the receptor, for instance, F152[4.60], Y103[3.37], and F250[6.55] make close contact with the acyl chain of LysoPS.

We next used functional assay to validate the observed interactions between receptor and ligand. The R75A[2.60], R156A[4.64], K98A[3.32], and Y22A[1.35] abolish or substantially reduce receptor activity ($\Delta pEC_{50}$,

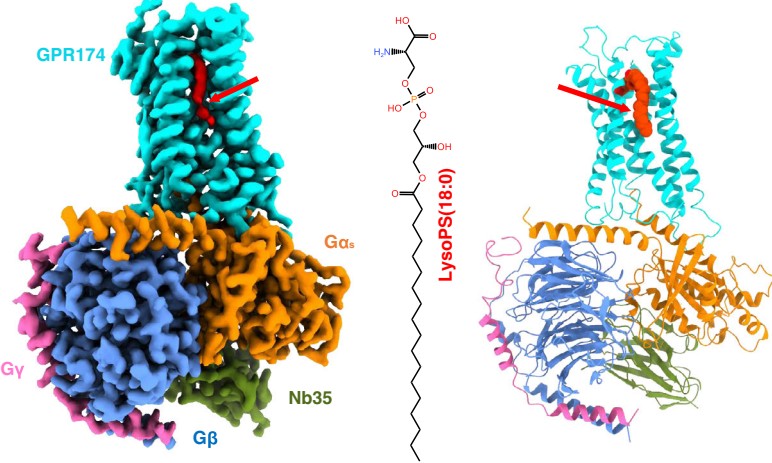

**Fig. 1 | The overall structure of GPR174/$G_s$ complex.** Left panel, orthogonal views of the cryo-EM density map of the GPR174/$G_s$ complex; the right panel, a model of the complex in the same view and color scheme as shown in the left panel.

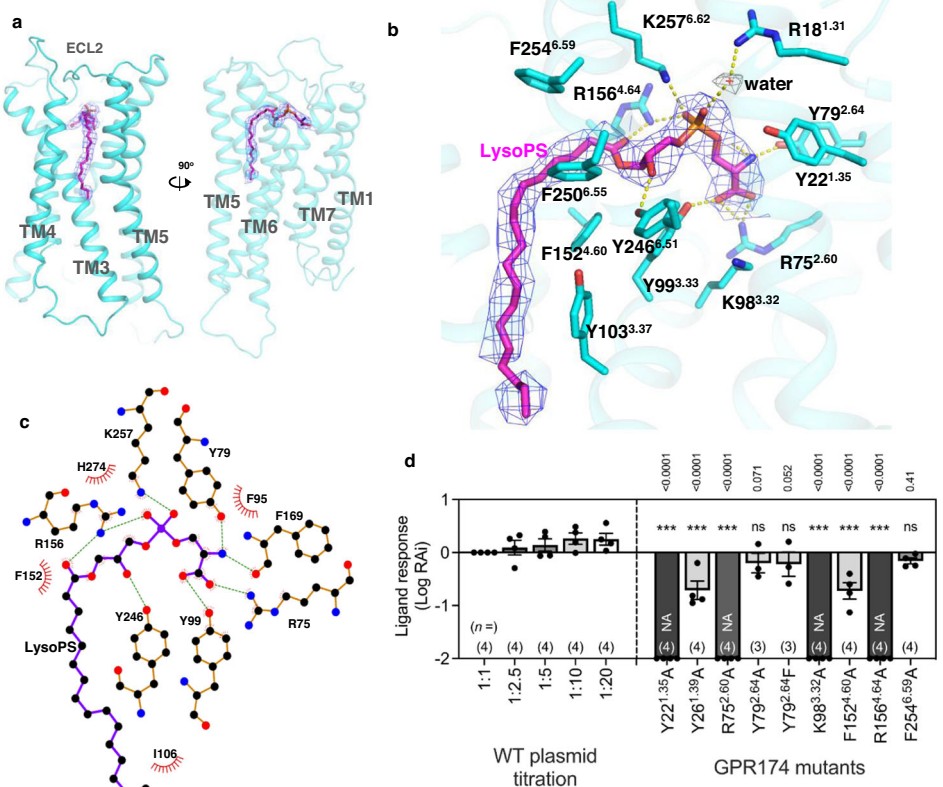

**Fig. 2 | Ligand binding pocket of GPR174. a** The overall structure of GPR174 with LysoPS ligand in it (magenta color). **b** An enlarged view of the ligand binding pocket focused on the head of LysoPS (magenta). Density map of the ligand (blue mesh) and water (gray mesh) is set at a contour level of 5.0 in pymol. **c** An interaction plot of LsyoPS head group and key residues of receptor–ligand binding pocket by LigPlot. Green line, polar interaction; eyelash shape, hydrophobic interaction. **d** Ligand response of GPR174 mutants ($\Delta$pEC$_{50}$ to WT GPR174). Data were plotted as mean values ± SEM. Numbers in the parentheses above the x-axis denote numbers of independent experiments with individual data points shown as dots. NA parameter not available owing to a lack of ligand response. Statistical analyses were performed using the ordinary one-way ANOVA followed by the two-sided Sidak's post hoc test with the expression-matched (colored) WT response. ns $p > 0.05$; **$p < 0.01$; ***$p < 0.001$.

Fig. 2d, expression level, and concentration-response; Supplementary Fig. 7), consistent with their roles to form key polar interactions with the polar head of LysoPS. The Y26A$^{1.39}$ only slightly reduces receptor activity, presumably due to the longer distance to LysoPS polar head. The Y79A$^{2.64}$ and Y79F$^{2.64}$ mutants show no effect on receptor activity probably due to the redundancy of this interaction to the existing extensive interactions of Y99$^{3.33}$, R75$^{2.60}$, F169$^{ECL2}$ to the L-serine group (Figs. 2b and 3b). The F254A$^{6.59}$ shows almost no effect on receptor activity, probably due to its distance from the ligand.

We also used molecular dynamic (MD) simulation to examine the binding mode of LysoPS in GPR174 (Supplementary Fig. 5a). In the three 500-ns trajectories, the LysoPS molecule fluctuated, but the acyl tail was mainly kept in the TM4–TM5 groove during the simulations as in the experimental observation (Supplementary Fig. 5b, c). The relative position of the non-acyl chain moiety, including the polar head group, tended to drift within the positively charged cavity (Supplementary Fig. 5d). Especially, the key R156$^{4.64}$ residue retained the direct hydrogen bond with LysoPS as implied by the experimental structure and mutant study. The other key residues identified by the mutant study (Y22$^{1.35}$, R75$^{2.60}$, and K98$^{3.32}$) showed weaker interactions than R156$^{4.64}$, suggesting their other roles in complicated ligand recognition.

## The receptor binding mode of LysoPS

Several types of lysophospholipids function as lipid mediators. All LysoGPs are zwitterionic molecules with a negatively charged head and a highly hydrophobic acyl tail, and many of them serve as signaling molecules to regulate a broad spectrum of

physiologies via binding to GPCRs. We, therefore, asked whether the receptor binding modes of LysoGPs are similar by comparing recently solved lysophospholipid receptors including S1P-bound S1PR1/G$_i$ complex[21], LPA-bound LPAR1/G$_i$ complex[17] and monoolein-bound zebrafish LPAR6a crystal structure[20]. A superimposition of these receptors with LysoPS-bound GPR174 shows that S1PR1 and LPAR1, both of which belong to the EDG lipid receptor family[22], have similar receptor binding modes, the phosphate heads of them stand up and point to the extracellular side, forming polar interactions with the conserved tyrosine and lysine of the N-terminal Helix (Fig. 3a, c, and d). On the other hand, GPR174 and LPAR6a, both of which belong to the non-EDG lipid receptor family, show a totally different receptor binding model: the acyl chains in the ligands are embedded into the cleft composed of TM3-5 (Fig. 3e). The cleft of the GPR174/G$_s$ complex is narrower than that of the LPAR6a crystal structure, suggesting the opened cleft for the lateral lipid access[23] closes in the presence of G proteins. Moreover, LysoPS-bound GPR174 shows clearly defined group interactions that are missing in the non-physiological monoolein-bound LPAR6a structure[20]. Instead of the standing-up head conformation in the EDG family receptors, the L-serine head group of LysoPS is buried deep into a positively charged cavity formed by R75$^{2.60}$, K98$^{3.32}$, Y99$^{3.33}$, and Y79$^{2.64}$ (Fig. 3a, b), making extensive polar interactions with these key residues and the backbone carbonyl group of F169$^{ECL2}$. In addition, the phenol ring of F169$^{ECL2}$ also forms cation-$\pi$ interaction with the positively charged R75$^{2.60}$ to further stabilize the network interaction of the L-serine head (Fig. 3b). Together with the

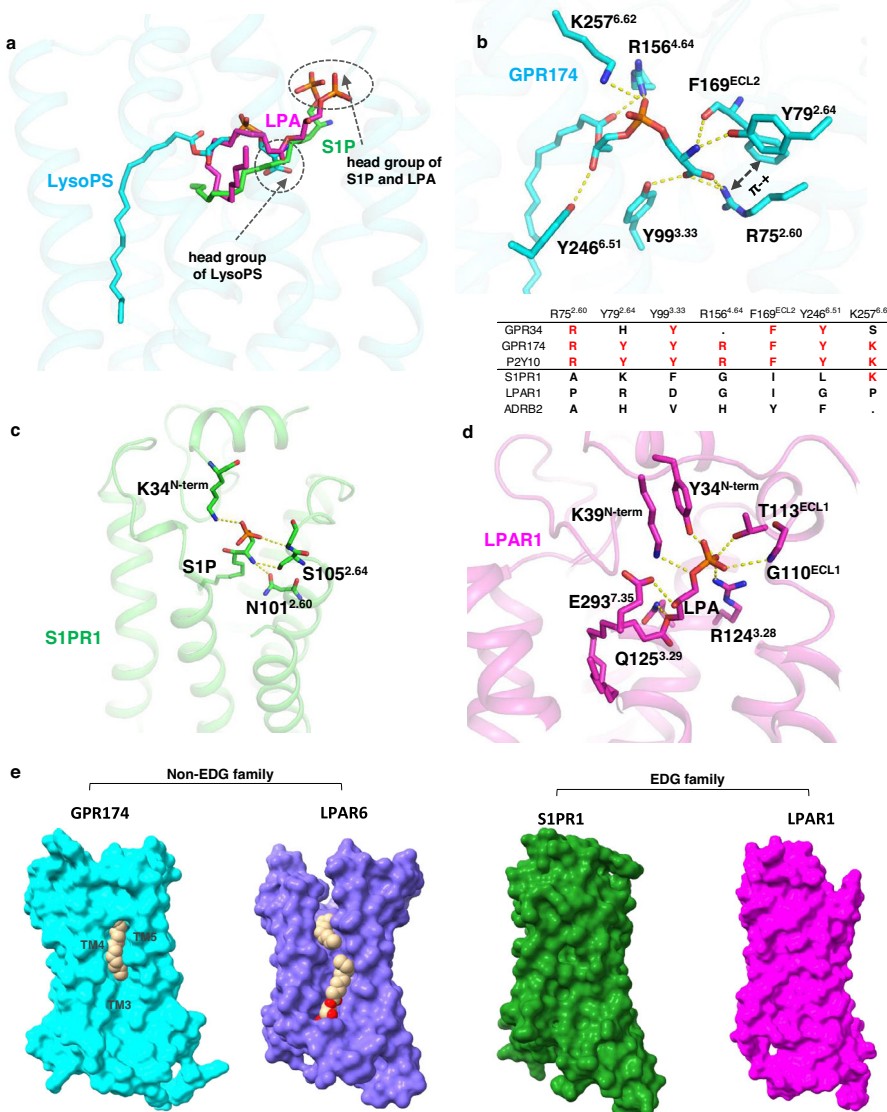

**Fig. 3 | A comparison of receptor binding of LysoPS with LPA and S1P. a** A superimposition of LysoPS with LPA (PDB:7td0) and S1P (PDB:7wf7) in GPR174. **b** The detail of polar interactions between LysoPS and the receptor in the ligand pocket. An alignment of key residues of the LysoPS family and S1PR1, LPAR1, and β2AR in the lower panel, the red color marks the conserved residues of LysoPS receptors. **c** A detail of polar interaction between the head group of S1P and S1PR1 (PDB:7vie). **d** A detail of polar interaction between the head group of LPA and LPAR1 (PDB:7td0). **e** A comparison of the side-entry of ligand binding pocket between no-EDG family and EDG family. LPAR6 (PDB:5xsz), S1PR1 (PDB:7wf7), and LPAR1 (PDB:7td0).

phosphate interaction of K257[6.62] and R156[4.64], and the interaction of Y246[6.51] on the *sn*−2 hydroxyl group of the glycerol group, form a massive polar interaction network to firmly lock the ligand into the pocket. An alignment of LysoPS receptor GPR34, GPR174, and P2Y10, with other LysoGPs receptor LPAR1 and S1PR1, as well as the G$_s$-coupled β2AR, show that the key residues that form the network polar interaction with LysoPS, R75[2.60], Y79[2.64], Y99[3.33], R156[4.64], F169[ECL2], Y246[6.51], and K257[6.62], are conserved in all of the three LysoPS receptors (GPR174, GPR34, and P2Y10), but not in S1PR1, LPAR1, and β2AR (Fig. 3b, lower panel), suggesting that LysoPS receptor GPR34 and P2Y10 may use the same recognition model for LysoPS binding.

We further used the docking method to examine whether LysoPS adopts a similar position in GPR34 and P2Y10. Since there are no GPR34 and P2Y10 structures available, we used AlphaFold predictions[24] of GPR34 and P2Y10 as the initial template, then modeled with Rosetta-Fold predictions[25] of GPR34 and P2Y10, as well as the active GPR174 (see the "Methods" section for details) for the docking. In docking analysis,

the top-scored docking positions of LysoPS molecules in GPR34 and P2Y10 superimposed well that of GPR174 (Supplementary Fig. 6a, b), supporting that the LysoPS receptors use a similar mode for LysoPS binding. Interestingly, a recent report suggested that acylation of the L-serine of LysoPS converts agonist to antagonist[26]. We used an induced-fit docking method to examine the possible role of the L-serine acylation of LysoPS on receptor binding, for this purpose, we docked compound 8a (cpd8a) of the LysoPS acylation paper[26] into the ligand binding pocket of GPR174 (Supplementary Fig. 6c). Comparing to LPS, the top scored cpd8a in the docking showed a loss of interactions with R75[2.60], Y79[2.64], and F169[ECL2], and a gain of interactions with Y22[L.35] and H274[7.35] via the by the acylation of the L-serine (Supplementary Fig. 6d). We speculated that the switch of binding mode induces a conformation to inhibit receptor activation.

## Activation of GPR174

Since there is no inactive GPR174 structure available, we compared the crystal structure of antagonist-bound P2Y1[24], the most closely related

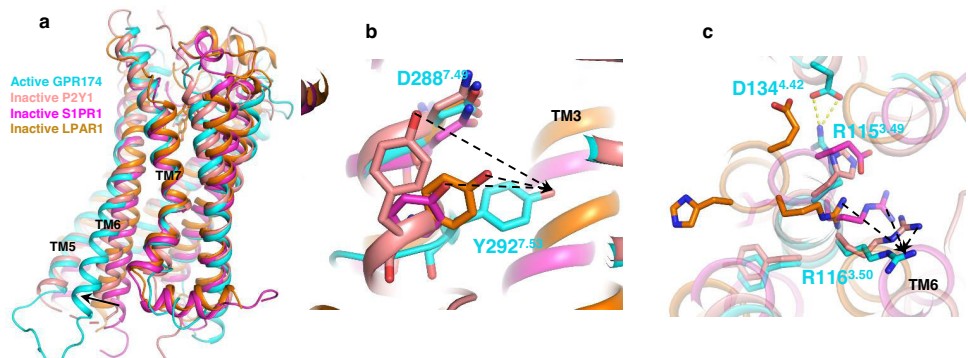

**Fig. 4 | Activation of GPR174. a** A comparison of active GPR174 with the inactive P2Y1 (PDB: 4xnw), ML056-bound S1PR1 (PDB:3v2y), and ONO-97808307-bound LPAR1 (PDB:4z34). **b** A comparison of the NPxxY motif between the active GPR174 and the inactive P2Y1, S1PR1, and LPAR1. **c** A comparison of the DRY motif between the active GPR174 and the inactive P2Y1, S1PR1, and LPAR1.

receptor that had been solved in its inactive state, as well as the crystal structures of ML056-bound S1PR1[27] and ONO-97808307-bound LPAR1[28], with LysoPS-bound GPR174. A superimposition of the active GPR174 on the above inactive receptors shows the most notable change is the outward displacement of TM6 in the active GPR174 (Fig. 4a), a signature of GPCR activation, allowing the αH5 of Gα to engage the receptor. We also compared the conserved NPxxY, DRY, and PIF motifs upon receptor activation, those motifs have been demonstrated to play crucial roles in receptor activation[29]. In the NPxxY motif, we observed a typical bending displacement of Y292[7.53] toward the space between TM3 and TM6 (Fig. 4b). GPR174 does not have a typical D[3.49]R[3.50]Y[3.51] motif, instead, the first D[3.49] is replaced with arginine. Interestingly, we found that R115[3.49] forms a salt bridge with D134[4.42], which locks and stabilizes ICL2 (Fig. 4c and Supplementary Fig. 8b). In the meantime, we also observed a typical displacement of R116 toward TM6. In the PIF motif, we did not observe much difference between the inactive P2Y1 and the active GPR174 (Supplementary Fig. 8a), which is not a surprise to us as the crystal structure of P2Y1 is in an intermediate state and the PIF motif may adopt to a partially active conformation.

### The Gₛ engagement of GPR174

We next looked at the Gₛ protein engagement of GPR174. To our surprise, we found the Gₛ engaging model of GPR174 is largely out of ordinary. First, we observed the most extensive polar interactions between receptor and Gαₛ, particularly the ICL3 end and the N-terminal of TM6, specifically, K225[6.30] interacts with Q384[G.H5.16] and D381[G.H5.13]; E224[6.29] interacts with R385[G.H5.17] of αH5 and Y358[G.S6.02] of β6 of Gαₛ; D221[6.26] also forms polar interaction with Y358[G.S6.02], and Q220[6.25] forms hydrogen interaction with the backbone carbonyl of C359[G.S6.01] of β6 of Gαₛ (Fig. 5a). We also compared the TM6/ICL3-Gα interaction of GPR174 with that of lipid receptors, including BLT1[30] and EP4[31]. We note that, due to the flexibility of this region, ICL3 structural information is missing in other lipid receptors such as S1PR1, LPAR1, and GPR119[32] (Supplementary Fig. 8d). A detailed comparison among GPR174, BLT1, and EP4 shows that GPR174 has the most extensive polar interaction between TM6/ICL3 and Gα protein, BLT1 has two cluster of interactions while EP4 only has one pair of polar interaction between TM6/ICL3 and Gα (Supplementary Fig. 8e–g). On the other side of the receptor, N297[8.48] form a hydrogen bond with E392[G.H5.24], D128[ICL2] forms polar interaction with Y391[G.H5.23] and the backbone carbonyl group of P123[ICL2] interacts with H387[G.H5.19] (Fig. 5c). Second, we noticed that the tip of αH5 is deeply inserted into the intracellular cavity of the receptor, making close contact with Y293[7.54] and M58[2.43] that are usually untouchable by other G proteins (Fig. 5b, c, and f).

A comparison of the Gₛ-engaging model of GPR174 with other Gₛ-coupling receptors shows that GPR174 uses a distinctive model to bind the αH5 of Gαₛ. Most Gₛ proteins engage to the TM5/TM6 side of the receptor, and the tips (C-termini) point to the TM6, this includes distinct Gₛ-coupled receptors: β2AR[33], melanocortin receptor 1 (MC1R)[19], GPR52[34], and parathyroid hormone receptor-1 (PTH1R)[35] (Fig. 5d). The Gₛ-coupled prostaglandin E receptor EP2[36] and EP4[31] resemble GPR174 where αH5 engages the central part of the receptor, however, the tips of αH5 of the EP2 (EP4)/Gₛ complexes all point to the outside of receptors, while in GPR174/Gₛ complex the tip of αH5 is buried deeply into the intracellular core of the receptor (Fig. 5e). In fact, the insert of αH5 tip into the intracellular core of GPR174 is the deepest insertion we ever observed in Gₛ-coupled receptors. Of particular interest, L393[G.H5.25] of αH5 inserts into a highly hydrophobic cavity formed by Y293[7.54], V55[2.40], F44[1.57], Y47[1.60], and F299[8.50] (Fig. 5f and Supplementary Fig. 8c), a phenomenon that has never been seen in other Gₛ-coupled receptors. Another interesting observation is the position of Y391[G.H5.23], in our structural comparison, Y391[G.H5.23] of different receptor/Gₛ complexes all point to the TM3, and usually form a hydrogen bond with the R[3.50] of the DRY motif as seen in the β2AR and many other class A GPCRs, however, Y391[G.H5.23] of the GPR174/Gₛ complex points to the ICL2 and TM1 direction and forms a hydrogen bond with D128[ICL2] (Fig. 5c, d), which has also never been seen in other Gₛ-coupled receptors.

### Discussion

In this study, we reveal the cryo-EM structure of LysoPS-bound GPR174 in a complex with Gₛ protein. Unlike LPA and S1P, LysoPS has a bulky, charged group (L-serine) on its head (Supplementary Fig. 9). The chemical structure difference suggests a different receptor binding mode for LysoPS than that for LPA and S1P. Consequently, LysoPS shows a receptor-engaging model, the L-serine head is buried deep into a positively charged pocket formed by R75[2.60], K98[3.32], Y26[1.39], Y22[1.35], Y79[2.64], and Y99[3.33] (Fig. 2b and Supplementary Fig. 4d). Most importantly, the key residues that make extensive network interactions with LysoPS are conserved in LysoPS receptor GPR34, GPR174, and P2Y10, but not in other LysoGP receptor S1PR1 or LPAR1 (Fig. 3b), indicating all LysoPS receptors use a similar model for ligand binding. The docking study of the N-acylation of LysoPS derivative suggested that agonists and antagonists use different sets of pocket residues for receptor binding (Supplementary Fig. 6d), the preference for using those residues holds a key to turning on or turning off receptors. Given the potential of developing an anti-GPR174 intervention for immune disorders treatments, our structure provides a rational basis for the design of antagonists of LysoPS receptors.

The distinctive Gₛ engagement of GPR174 is totally out of our expectations. The extensive polar interactions between receptor/Gαₛ and the deepest insertion of αH5 into the intracellular core of the receptor suggest that GPR174 uses a unique way to bind and engage

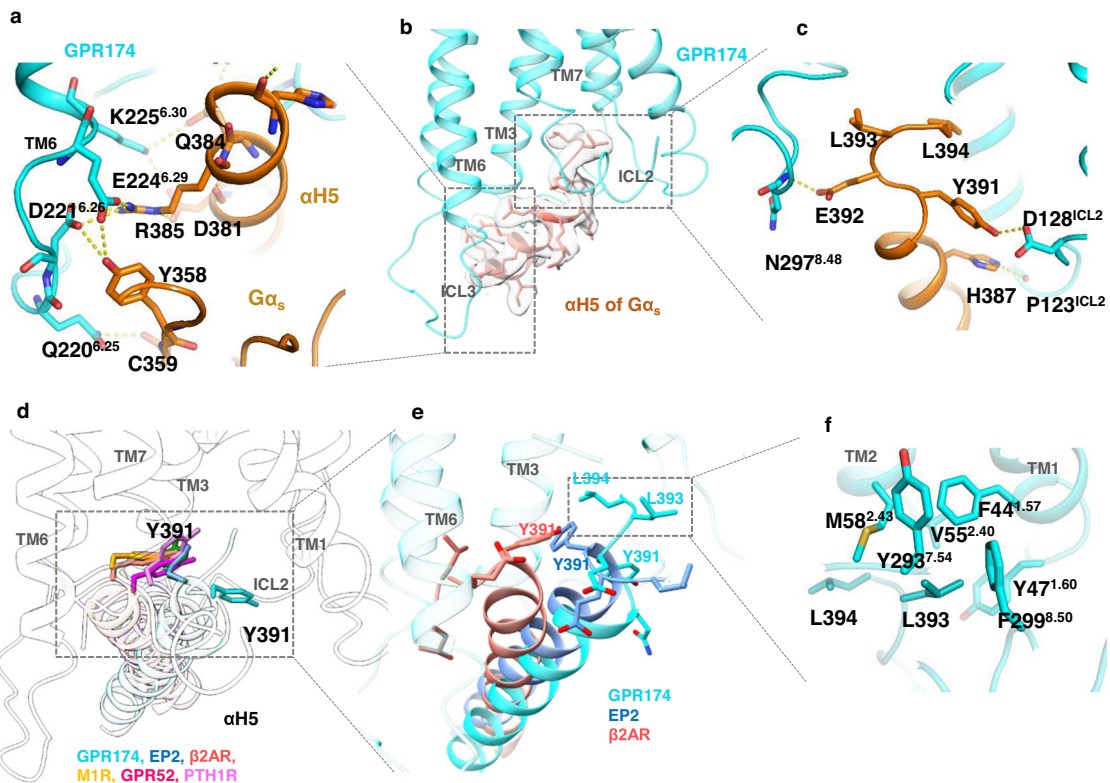

**Fig. 5 | The $G_s$ engaging model of GPR174. a** TM6/ICL3 junction forms extensive polar interaction with $G\alpha_s$. **b** The overall engagement of αH5. Density map was drawn by chimeraX at a map level of 0.08. **c** The detail of the αH5 head interaction. **d** A superimposition of αH5 engagements in the context of GPR174. **e** A detailed comparison of αH5 engagements among GPR174, EP2 (PDB: 7cx2) and β2AR (PDB: 3ns6). **f** L393 of the aH5 head inserts into hydrophobic pockets formed by residues from TM1,2,7 and H8.

$G\alpha_s$, and those observations also indicate that $G_s$ is tightly packed against receptor, and this may explain the high level of constitutive activity of GPR174[11,37] and weak ligand-induced $G_s$ dissociation activity (see the section "Methods" functional assay). We used the NanoBiT strategy in our complex assembling, the strategy only increases the local concentration of binding partners and cannot force an interaction that does not exist in nature, evidenced by the identical structures of GPR110/$G_s$ complexes solved with or without the NanoBiT strategy recently[38,39]. Taken together, our structure of the LysoPS-bound GPR174/$G_s$ complex provides a foundation to understand LysoPS signaling and clues for developing LysoPS intervention.

## Methods
### Constructs
The code-optimized human GPR174 (1–326) was cloned into pFastBac plasmid with haemagglutinin (HA) signal peptide at the N-terminus, and its C-terminus was fused with a LgBiT[18] followed by a Tobacco etch virus (TEV) cutting site and 2X fused maltose-binding proteins (MBP) to promote protein expression and purification. The HiBiT fusion of human Gβ and Gγ was described as before[40], the mini-$G\alpha_s$ was adopted from the MC1R/$G_s$ complex[19].

### Expression and purification of GPR174/$G_s$ complexes
Recombinant baculovirus encoding GPR174-LgBiT-TEV-2MBP, $G\alpha_s$, Gβ₁, and Gγ₂ proteins were co-expressed via infection in Spodoptera frugiperda (Sf9, Invitrogen, #11496-015) cells. Cells were cultured at 27 °C, 110 rpm for 48 h, at a density of $2 \times 10^6$ cells per ml before infection at a ratio of 1:100 (virus volume versus cell volume). Two days later, cells were collected and resuspended in lysis buffer (20 mM HEPES buffer, 150 mM NaCl, 10 mM MgCl₂, 20 mM KCl and 5 mM CaCl₂, pH 7.5). In order to improve the stability and expression of the GPR174–$G_s$ complex prepared in membranes, the mixture was incubated for 1.5 h at room temperature by adding apyrase (25 mU/ml), 1-stearyl LysoPS (1 μM, Avanti #858144), Nb35 (10 mg/ml) and after douncing ~30 times. Then, the sample was solubilized in the buffer of 0.5% (wt/vol) lauryl maltose neopentylglycol (LMNG; Anatrace) and 0.1% (wt/vol) cholesteryl hemisuccinate Tris salt (CHS) for 2 h at 4 °C and ultracentrifuged at 45,000 rpm ($64,000 \times g$) at 4 °C for 45 min to collect the supernatant. Complex was loaded on the amylose column for 2 h and washed with a buffer solution containing 1 μM LysoPS, 25 mM HEPES, pH 7.5, 150 mM NaCl, and 0.01% LMNG/0.002% CHS. Then the complex was eluted with the same buffer plus 10 mM maltose. After concentration and TEV cutting overnight at 4 °C, the complex protein was subjected to a Superdex 200 Increase 10/300 GL (GE Health Sciences) gel infiltration column pre-equilibrated with 1 μM LysoPS, 25 mM HEPES, pH 7.5, 150 mM NaCl and 0.00075% (wt/vol) LMNG, 0.00025% glyco-diosgenin (GDN), 0.0002% CHS (wt/vol) (Anatrace). The complex fractions were concentrated to 10 mg/ml and snap-frozen for later grid preparation.

### Expression and purification of Nb35
Nb35 was purified as described before[19], briefly, BL21 cells containing Nb35 plasmid were cultured in TB media supplemented adding to 0.1% glucose, 1 mM MgCl₂, and 50 μg ml⁻¹ kanamycin at 37 °C until OD 600 reaching 0.7. The cultures were induced by 1 mM IPTG, and cultured at 28 °C for 4 h. The cells were harvested by centrifugation at $3000 \times g$ for 20 min. Then, the precipitate was lysed in TES buffer of 0.2 M Tris, pH 8 and 0.5 mM EDTA, 0.5 M Sucrose by stirring at 4 °C for 3 h. The homogenate was ultracentrifuged at $10,000 \times g$ at 4 °C for 45 min. The supernatant was then incubated with a nickel column for 2 h at 4 °C. The sample was then washed with 20 column volumes of washing buffer (20 mM HEPES, pH 7.5, 150 mM NaCl) and washed again with

high salt buffer (43 mM $NaH_2PO_4 \cdot H_2O$, 7 mM $NaH_2PO_4$, pH 6 and 1 M NaCl). The Nb35 protein was eluted by 10 column volumes of 19 mM $NaH_2PO_4 \cdot H_2O$, 38 mM $NaH_2PO_4$, pH 6 and 150 mM NaCl, and 250 mM imidazole, and then loaded onto a Superdex 200 Increase 10/300 GL (GE Health Sciences) gel infiltration column by using 20 mM HEPES, pH 7.5 and 150 mM NaCl. The purified protein was concentrated to 10 mg ml$^{-1}$ with a 10-kDa molecular weight cut-off concentrator (Millipore), and then supplemented with 10% glycerol and stored at −80 °C until use.

## Grid preparation and cryo-EM data collection
A 3 μl receptor/G-protein complex sample (-10 mg/ml) was applied to a glow-charged quantifoil R1.2/1.3 Cu holey carbon grids (Quantifoil GmbH). The grids were vitrified in liquid ethane on a Vitrobot Mark IV (Thermo Fisher Scientific) instrument at the setting of blot force of 10, blot time of 5 s, humidity of 100%, and temperature of 4 °C. Grids was first screened on an FEI 200 kV Arctica transmission electron microscope (TEM) and then grids with evenly distributed particles in thin ice were transferred to an FEI 300 kV Titan Krios TEM equipped with a Gatan Quantum energy filter. Images were taken by a Gatan K2 direct electron detector at the magnitude of 64,000, a super-resolution counting model at a pixel size of 0.55 Å, and the energy filter slit was set to 20 eV. Each image was dose-fractionated in 40 frames using a total exposure time of 7.3 s at a dose rate of 1.5 e/Å$^2$/s (total dose 60 e/Å$^2$). All image stacks were collected by the FEI EPU program, nominal defocus value varied from −1.2 to −2.2 μm.

## Data processing
We use a similar pipeline to process data as described before[16]. A total of 2001 raw movies (0.55 Å) were binned (1.1 Å) and motion-corrected using MotionCor2[41], followed by CTF estimation by CTFFIND 4.1[42]. Particles (-1.5 million) were picked by crYOLO[43] and extracted by RELION[44] (version 3.1) and subjected to reference-free 2D classification in RELION. Good classes (- 700,000 particles) of well-defined features were passed to the next round for initial model generation and 3D classification. The initial model was generated by cryoSPARC[45] ab initio. The model was used as a reference in RELION 3D classification (-5 classes). The best class (-500,000) that showed clear secondary structure features was selected for a 3D refinement in RELION, followed by a Bayesian polishing[46], then a 3D refinement and a CTF refinement in RELION. The refined particles were subjected to a second round of 3D classification (3–4 classed) with a mask on the complex to yield a class of about 400,000 particles for final refinement by the cryoSPARC Non-uniform Refinement, which generated a map of 2.76 Å, based on the gold standard Fourier shell correlation (FSC) = 0.143 criterion. Local resolution estimations were performed using an implemented program in cryoSPARC.

## Model building
The AlphaFold[24] prediction of mouse GPR174 (AF-Q9BXC1-F1) and $G_s$ protein complex from MC1R (PDB 7f4d)[19] were used as initial models for model rebuilding and refinement against the electron microscopy map. All models were docked into the electron microscopy density map using UCSF Chimera[47] and then subjected to iterative manual adjustment in Coot[48], followed by a rosetta cryoEM refinement[49] at relax model and Phenix real space refinement[50]. The model statistics were validated using MolProbity[51]. Structural Figures were prepared in UCSF Chimera, ChimeraX[52], and PyMOL (https://pymol.org/2/).

## Structure and sequence comparison
Sequence alignment by the Clustal Omega[53] sever and the representation of sequence alignment was generated using the ESPript[54] website (http://espript.ibcp.fr). The generic residue numbering of GPCR is based on the GPCRdb[55] (https://gpcrdb.org/). The ligand/receptor interaction was plotted by the LigPlot[56].

## NanoBiT-$G_s$-coupling assay
LysoPS-induced $G_s$ activation was measured by a modified version of the NanoBiT-G-protein dissociation assay[10]. Since the $G_s$ dissociation response was too weak to be accurately measured, we instead measured $G_s$ coupling to GPR174 using the NanoBiT enzyme complementation system (Promega), in which we fused the small and the large luciferase fragments to the C-terminus of GPR174 and the helical domain of the $G\alpha_s$ subunit, respectively. HEK293A cells (Thermo Fisher Scientific, #R70507) were seeded in a 6-well culture plate (Greiner Bio-One) at a concentration of $2 \times 10^5$ cells per ml (2 ml per well hereafter) in DMEM (Nissui Pharmaceutical) supplemented with 5% FBS (Gibco), glutamine, penicillin, and streptomycin, one day before transfection. Transfection solution was prepared by combining 6 μl of polyethylenimine solution (1 mg per ml) and a plasmid mixture consisting of 500 ng LgBiT-containing $G\alpha_s$, 500 ng $G\beta_1$, 500 ng $G\gamma_2$, 100 ng RIC8A (G-protein chaperon), and 500 ng of the GPR174 construct (N-terminal hemagglutinin signal sequence followed by a FLAG epitope tag, plus C-terminal SmBiT with a 15-amino acid flexible linker). After incubation for one day, the transfected cells were harvested with 0.5 mM EDTA-containing Dulbecco's PBS, centrifuged, and suspended in 2 ml of Hank's balanced saline solution (HBSS) containing 0.01% bovine serum albumin (BSA fatty acid–free grade, SERVA) and 5 mM HEPES (pH 7.4) (assay buffer). The cell suspension was dispensed in a white 96-well plate at a volume of 80 μl per well and loaded with 20 μl of 50 μM coelenterazine (Amadis Chemical), diluted in the assay buffer. After 2 h incubation, the plate was measured for baseline luminescence (SpectraMax L, Molecular Devices), and 20 μl of 6 × 1-oleoyl-LysoPS (Avanti Polar Lipids), serially diluted in the assay buffer, were manually added. The plate was immediately read for the second measurement as a kinetics mode and luminescence counts recorded from 5 to 10 min after compound addition were averaged and normalized to the initial counts. The fold-change signals were further normalized to the vehicle-treated signal and were plotted as a G-protein-coupling response. Using the Prism 9 software (GraphPad Prism), the G-protein activation signals were fitted to a four-parameter sigmoidal concentration–response curve and obtained $EC_{50}$ and Span ("Top"–"Bottom") values. For individual experiments, we calculated Span/$EC_{50}$ of GPR174 mutants relative to that of WT GPR174, a dimensionless parameter known as relative intrinsic activity (RAi)[57], and used its log-transformed value (Log RAi) to denote ligand response activity of the mutants.

## Flow cytometry
Transfection was performed according to the same procedure as described in the "NanoBiT-$G_s$-coupling assay" section. One day after transfection, the cells were collected by adding 200 μl of 0.53 mM EDTA-containing Dulbecco's PBS (D-PBS), followed by 200 μl of 5 mM HEPES (pH 7.4)-containing Hank's Balanced Salt Solution (HBSS). The cell suspension was transferred to a 96-well V-bottom plate in duplicate and fluorescently labeled with an anti-FLAG epitope (DYKDDDDK) tag monoclonal antibody (Clone 1E6, FujiFilm Wako Pure Chemicals #012-22384; 10 μg per ml diluted in 2% goat serum- and 2 mM EDTA-containing D-PBS (blocking buffer), 1:200) and a goat anti-mouse IgG secondary antibody conjugated with Alexa Fluor 488 (Thermo Fisher Scientific, 10 μg/ml diluted in the blocking buffer). After washing with D-PBS, the cells were resuspended in 200 μl of 2 mM EDTA-containing-D-PBS and filtered through a 40-μm filter. The fluorescent intensity of single cells was quantified by an EC800 flow cytometer equipped with

a 488 nm laser (Sony). The fluorescent signal derived from Alexa Fluor 488 was recorded in an FL1 channel, and the flow cytometry data were analyzed with the FlowJo software (FlowJo). Live cells were gated with a forward scatter (FS-Peak-Lin) cutoff at the 390 settings, with a gain value of 1.7. Values of mean fluorescence intensity (MFI) from approximately 20,000 cells per sample were used for analysis. Typically, we obtained a WT MFI value of ~1600 (arbitrary unit) and a mock MFI value of ~20. For each experiment, we normalized an MFI value of the mutants by that of WT performed in parallel and denoted relative levels.

## MD simulation

**System preparation.** The receptor coordinates were extracted from the experimental structure (residues 11-303), and the missing atoms were fulfilled with the program Modeller 9.25[58] for CHARMM-GUI Membrane Builder[59–62]. The receptor was patched with neutral acetyl and methylamide groups at the terminal residues, and titratable residues were kept at their dominant states under neutral conditions, except for D65$^{2.50}$, as in other active-state GPCRs. The system was then embedded in a palmitoyloleoylphosphatidylcholine (POPC) bilayer, solvated with explicit TIP3P waters and neutralized by adding 150 mM NaCl. The resulting periodic boundary system contained a total of 105,654 atoms in a box of about $96 \times 96 \times 125$ Å³ ($X \times Y \times Z$, $X–Y$ plane as the membrane plane). CHARMM36m force field[63] and CHARMM General Force Field[64] were used for the following calculation.

**Simulation.** Simulations were performed by CHARMM-GUI standard protocols[61] using GROMACS 2020.4[65]. Minimization was performed until the maximum force dropped below 1000 kJ mol⁻¹ nm⁻² with heavy atom restraints ((i) 4000 kJ mol⁻¹ nm⁻² for main chain and ligand position; (ii) 2000 kJ mol⁻¹ nm⁻² for side chain position; (iii) 1000 kJ mol⁻¹ nm⁻² for $Z$ position of the phosphorus atom in POPC; (iv) 1000 kJ mol⁻¹ rad⁻² for the dihedral angle of POPC chiral carbon (reference angle of −120° and threshold angle of 2.5°) and the double bond (reference angle of 0° and threshold angle of 0°)).

Equilibration was performed by gradually decreasing the restraints under the NVT and NPT conditions. Temperature and semi-isotropic pressure were maintained at 310 K and 1.0 bar using a Berendsen thermostat, and long-range electrostatics were calculated using the particle mesh Ewald method[66]. NVT equilibration was started with the same restraints in the minimization step for 125,000 steps with 1 fs time step, followed by further 125,000 steps with decreased restraints ((i) and (ii) 50% of the minimization step; (iii) and (iv) 40%). NPT equilibrations were performed for 125,000 steps with restraints ((i) and (ii) 25%; (iii) 40%; (iv) 20%), 250,000 steps of 2 fs time step with restraints ((i) 12.5%; (ii) 10%; (iii) and (iv) 20%), 250,000 steps with restraints ((i) 5%; (ii) 2.5%; (iii) 4%; (iv) 10%), and 250,000 steps with restraints ((i) 1.25%; (ii)–(iv) 0%).

Production run was performed with 2 fs time step under the NPT condition of 1.0 bar and 310 K by using Nose–Hoover method[67] and Parrinello–Rahman method[68].

**Analysis.** Trajectories were manipulated with CPPTRAJ[69] and then analyzed and visualized using MDTraj[70], Matplotlib[71], and PyMOL (Schrödinger). For Supplementary Fig. 5c, molecular snapshots for every 1 ns were aligned with Cα atoms and the coordinates of non-acyl chain moiety in LysoPS were used for RMSD calculation. The values were then subjected to hierarchical clustering implemented in SciPy[72] and clustered with a threshold of 8 Å. Centroidal coordinates were extracted by considering all heavy atoms.

## Molecular docking

The docking method is similar to the previous study[30]. Briefly, the AlphaFold[24] prediction of GPR34 (AF-Q9UPC5) and P2Y10 (AF-

O00398) were used as an initial template, then modeled with RosettaFold prediction[25] of active GPR34 and P2Y10, as well as the active GPR174 by modeller 10.0[73], then the active models of GPR34 and P2Y10 were prepared and minimized as before, the charged status of the pocket residues was determined by Protonate3D[74] of the MOE package. The LysoPS was placed in the ligand binding pocket using the triangle matcher with a London docking score. Refinement was employed based on a rigid receptor and GBVI/WSA docking scoring. The top-scored poses and statistics were shown in Supplementary Fig. 6. For the N-acylation of LysoPS derivative cpd8a docking, we used the cryo-EM structure of active GPR174 as the start model, the receptor was prepared and minimized as above. Refinement was employed based on the induced fit model and GBVI/WSA docking scoring.

## Reporting summary

Further information on research design is available in the Nature Portfolio Reporting Summary linked to this article.

## Data availability

The data that support this study are available from the corresponding authors upon reasonable request. The cryo-EM density maps have been deposited in the Electron Microscopy Data Bank (EMDB) under accession code EMD-33479 (GPR174/G$_s$ complex), and the coordinates have been deposited in the Protein Data Bank (PDB) under accession number 7XV3 (GPR174/G$_s$ complex). The mdp file for the MD simulation, the topology file of LysoPS, and.pdb file for the system were deposited to Zenodo [https://doi.org/10.5281/zenodo.7364300]. Previously published PDBs can be access via accession codes 7F4D, 7TD0, 7WF7, 7VIE, 5XSZ, 4XNW, 3VZY, 4Z34, 7CX2, 3NS6. The source data underlying Fig. 2d, and Supplementary Fig. S7a, b are provided as a Source Data file. Source data are provided with this paper.

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

## Acknowledgements

We thank Kayo Sato, Shigeko Nakano, and Ayumi Inoue at Tohoku University for their assistance in plasmid preparation and cell-based GPCR assays. This work was supported by the National Natural Science Foundation of China (32070048 to Y.H.). A.I. was funded by the Japan Society for the Promotion of Science (JSPS) KAKENHI grants JP21H04791, JP21H05113, JPJSBP120213501, and JPJSBP120218801; FOREST Program JPMJFR215T and JST Moonshot Research and Development Program JPMJMS2023 from Japan Science and Technology Agency (JST); JP22ama121038 and JP22zf0127007 from Japan Agency for Medical Research and Development (AMED); The Uehara Memorial Foundation; and Daiichi Sankyo Foundation of Life Science. The MD simulation was carried out using the TSUBAME 3.0 supercomputer at the Tokyo Institute of Technology.

## Author contributions

J.L. made the expression constructs, purified the proteins, prepared and screened the grids, analyzed data, and prepared figures. A.I. designed experiments, performed functional assays, analyzed data, and edited the manuscript. T.I. analyzed data, performed MD simulations, prepared figures, and edited the manuscript. R.X., N.W., K.K., Z.X., Y.Q., and X.Z. set the experiments, prepared plasmids, and cultured cells. A.Z. and C.G set cryo-EM and collected data. Z.H. analyzed data and edited the manuscript. Y.H. conceived the project, designed experiments, analyzed data, solved the structures, wrote the manuscript, and supervised the project. All authors contributed to the data interpretation and preparation of the manuscript.

## Competing interests

The authors declare no competing interests.
