## [Peer Review File · Nature Communications]

Structural basis of lysophosphatidylserine receptor GPR174
ligand recognition and activationReviewers' Comments:

Reviewer #1:

Remarks to the Author:

In this paper, He and colleagues characterized the structure of GPR174 in complex with LysoPS and a stimulatory G protein, Gs by cryoEM. They compared the GPR structure with other lipid binding GPCRs, including S1PR and LPA1R. The GPR174 structure reveals unique binding mode for the head group of phosphatidylserine, which is distinct from the headgroups of phosphate acid in LPA and sphingosine-1-phosphate in S1P. The structure of GPR174 also reveals a distinct Gs coupling from other Gs coupled GPCRs, suggesting diverse mode of GPCRs to couple with the same subtype of G proteins. Overall the structure is very well done and is of high quality. Given the importance of GPR174 in T cell biology and autoimmune diseases, the structure works presented in this paper could facilitate drug discovery targeting this interesting receptor. Below are minor points for revision.

1. The binding mode of LysoPS in GPR174 should be confirmed by MD simulation, which would indicate whether the binding mode is in a low energy and stable state.
2. P2Y10 and GPR34 are in the same subfamily as GPR174, and they contain similar amino acids that bind to the head group of LysoPS. Docking studies need to be performed with the AlphaFold model of P2Y10 and GPR34, and compared these docking models with the GPR174 structures.
3. Line 87, M1R should be spelled as MC1R.
4. Line 234, the sentence need to be rephrase to better english.

Reviewer #2:

Remarks to the Author:

Although several structures of GPCR, those ligand is a lipid, have been reported already, the number of examples is still limited. This work by Liang and Inoue et al provides a new addition to the example. This study is the first report of the structure of lysophosphatidylserine receptors. Lysophosphatidylserine receptors are probably important but the physiological functions are still unclear. The present result clearly showed that the hydrophilic serine head group was recognized by the receptor, human-GPR174. The interaction diagrams between the ligand and the receptor were supported by the mutant experiments. The authors compared the ligand binding poses among other lipid GPCRs, such as LPAR6, LPA, and S1P. Similarity and difference are clearly shown. This is interesting in terms of evolution of the GPCR system and probably in terms of ligand design in medicinal chemistry. I agree with publication of this work in Nature Communications. Before publication, the authors need to add a brief comment in the main text about the following two issues

- 1) the mutation result of K98: while the mutation experiment showed K98A mutation lost the ligand response completely, Fig 2(b) does not show any interaction of K98 with the ligand. Does this mutation change the receptor properties (localization) or the shape dramatically?
- 2) The authors discussed the interaction of TM6/ICL3 with G-protein alpha among several non-lipid GPCRs (Figure 5): how much this kind interaction general among lipid-ligand GPCRs?

REVIEWER COMMENTS

Reviewer #1 (Remarks to the Author):

In this paper, He and colleagues characterized the structure of GPR174 in complex with LysoPS and a stimulatory G protein, Gs by cryoEM. They compared the GPR structure with other lipid binding GPCRs, including SIPR and LPA1R. The GPR174 structure reveals unique binding mode for the head group of phosphatidylserine, which is distinct from the headgroups of phosphate acid in LPA and sphingosine-1-phosphate in SIP. The structure of GPR174 also reveals a distinct Gs coupling from other Gs coupled GPCRs, suggesting diverse mode of GPCRs to couple with the same subtype of G proteins. Overall the structure is very well done and is of high quality. Given the importance of GPR174 in T cell biology and autoimmune diseases, the structure works presented in this paper could facilitate drug discovery targeting this interesting receptor. Below are minor points for revision.

Thank you for your positive comment on our study.

1. The binding mode of LysoPS in GPR174 should be confirmed by MD simulation, which would indicate whether the binding mode is in a low energy and stable state.

Thank you for your suggestion. We performed MD simulation on the LysoPS-bound GPR174 (in the absence of Gs). The data show that the LysoPS position of the cryo-EM structure is one of the most stable positions. We also saw that the key residues identified by the mutant study (R75, K98 and R156) are all involved in ligand recognition (Fig. 1 of rebuttal letter and Supplementary Fig. 5a of the revision manuscript).

Rebuttal Fig. 1. MD simulation of LysoPS-bound GPR174.

2. *P2Y10* and *GPR34* are in the same subfamily as *GPR174*, and they contain similar amino acids that bind to the head group of LysoPS. Docking studies need to be performed with the AlphaFold model of *P2Y10* and *GPR34*, and compared these docking models with the *GPR174* structures.

Thank you for your suggestion. We applied docking to the AlphaFold model of *GPR34* (AF-Q9UPC5) and *P2Y10* (AF-O00398), the data shows that the top 1 scored docking pose is very similar to the LysoPS pose in *GPR174* (Fig. 2 of rebuttal letter and Supplementary Fig. 5b).

Rebuttal Fig. 2. Docking of LysoPS into GPR34 and P2Y10.

3. Line 87, *MIR* should be spelled as *MC1R*.

Corrected.

4. Line 234, the sentence need to be rephrase to better english.

Corrected, thank you. The corrected sentence now reads (edits are underlined):

“The chemical structure difference suggests a different receptor binding mode for LysoPS than that for LPA and S1P.” (Line 254-255)

Reviewer #2 (Remarks to the Author):

Although several structures of GPCR, those ligand is a lipid, have been reported already, the number of examples is still limited. This work by Liang and Inoue et al provides a new addition to the example. This study is the first report of the structure of lysophosphatidylseine receptors. Lysophosphatidylseine receptors are probably important but the physiological functions are still unclear. The present result clearly showed that the hydrophilic serine head group was recognized by the receptor, human-GPR174. The interaction diagrams between the ligand and the receptor were supported by the mutant experiments. The authors compared the ligand binding poses among other lipid GPCRs, such as LPAR6, LPA, and S1P. Similarity and difference are clearly shown. This is interesting in terms of evolution of the GPCR system and probably in terms of ligand design in medicinal chemistry.

Thank you for your positive comments on our study.

I agree with publication of this work in Nature Communications. Before publication, the authors need to add a brief comment in the main text about the following two issues

1) the mutation result of K98: while the mutation experiment showed K98A mutation lost the ligand response completely, Fig 2(b) does not show any interaction of K98 with the ligand. Does this mutation change the receptor properties (localization) or the shape dramatically?

Thank you for pointing this out. Indeed, K98 did not make a direct contact with LysoPS in the cryoEM structure (Fig. 2c). Yet, it is located in close proximity of LysoPS, we performed a mutagenesis study at the residue. By following the Referee #1's comment, we performed an MD simulation and observed occurrence of a contact between K98 and the polar head group (carboxyl moiety) of LysoPS (Fig. 1 of rebuttal letter). Thus, we consider that LysoPS molecule can adopt several semi-stable configurations within the ligand binding pocket and one of them involves K98 as a direct recognition.

The MD result and related texts are included in Supplementary Fig. 5a and Line 144-151, respectively, in the revision manuscript.

2) The authors discussed the interaction of TM6/ICL3 with G-protein alpha among several non-lipid GPCRs (Figure 5): how much this kind interaction general among lipid-ligand GPCRs?

Thank you for your suggestion. We compared with all available lipid receptor TM6/ICL3 interactions with $G\alpha$ (Fig. 3a of rebuttal letter). Except for BLT1 and EP4, all of the structurally determined lipid receptors do not have an intact ICL3 because of the flexibility of the region. Regardless, we compared them in detail (Fig. 3b-d and Supplementary Fig. 7e-g) and found that GPR174 has the most extensive polar interaction between TM6/ICL3 and $G\alpha$ protein, BLT1 has two cluster of interactions while EP4 only have one pair of polar interaction between TM6/ICL3 and $G\alpha$.

The TM6/IC3- $G\alpha$ comparison and related texts are included in Supplementary Fig. 7d and Line 219-226, respectively, in the revision manuscript.

Rebuttal Fig. 3. Comparisons of lipid receptor TM6/ICL3 interaction with $G\alpha$.

Reviewers' Comments:

Reviewer #1:

Remarks to the Author:

This is a very interesting paper, and the authors have addressed all my concerns in the first review, I would thus support its publication.

Reviewer #2:

Remarks to the Author:

This is the revised version of the manuscript which I reviewed. The authors clearly addressed to my queries about the missing interaction of K98 with LysoPS, and a character of the interaction of TM6/ICL3 with Galpha protein. I believe these additions of the discussions make this work much more scientifically rigorous. I agree with publication of this work in Nature Communications without further change.

Reviewer #3:

Remarks to the Author:

In the paper "Structural basis of lysophosphatidylserine receptor GPR174 ligand recognition and activation" by Liang et al., one good resolved cryo-EM structure of GPR173/GS complex bound with LysoPS (18:0) was presented using method called NanoBiT tethering strategy. The overall resolution of $\sim 2.8\text{\AA}$ was good enough to capture those crucial interactions between GPCR and ligand (lysoPS) as well as with other G proteins. The unique ligand binding mode and spatial engagement of GS protein within the complex were discussed in this manuscript decently. Molecular dynamics (MD) simulations of GPCR/lysoPS complex in POPC bilayer were conducted based on the suggestions from the last round of peer-review. Overall, my opinion of this manuscript is that it's acceptable to be published in Nature Communications after some examinations of the data are taken carefully to support the arguments they proposed. Here are my major concerns and minor suggestions.

Major concerns:

1. The density map of the cryo-EM structure of Sphingosine 1-phosphate receptor 1 / Gi complex bound to S1P (PDB ID: 7WF7) is clearly not supporting the model the authors presented in Figure 3.c in terms of the orientation of side chains of all of those residues in the panel. The authors should refine model of 7WF7 based on the map to obtain more sensible binding pose and interactions otherwise the comparison and discussion are not correct.
2. In the section of discussing the activation mechanism of GPR174, the authors using a group of comparisons between the active form of GPR174 and an inactive form of P2Y1. However, how closely these two receptors are related? More importantly, comparing a cryo-EM structure solved using single-particle technique with a crystal structure with heavy crystal packing is tricky, one sample of related receptor with antagonist binding is also not enough. More structures shall be involved in this discussion.
3. Still about the inactivation form, since the authors have already noticed that the acylation of the L-serine of LysoPS will antagonize the receptor, had the authors tried to solve the structure of GPR174 with the acylated LysoPS?
4. The most of my concerns come from the MD part, the authors conducted MD simulations with three replicas to study if the binding mode of ligand in GPR174 is stable and energetic favorable. However, the authors didn't discuss this topic sufficiently, no energy calculation was conducted but three RMSD

profiles of LysoPS. More importantly, several interactions observed in cryo-EM structure was lost in the initial model they used in MD simulation, for example, the interaction between Y79 and amine-group of L-serine in head group, as well as the interaction F169 and that amine group, which is due to the change of the position of L-serine in pocket. The authors should run proper equilibration before the production MD to at least maintain the binding pose first then discuss the stability of ligand.

5. The tail of LysoPS was shown to be flexible which makes sense to me since it would partially interact with the bilayer lipids according to the position of LysoPS in Figure 1, however, the authors didn't report the meaningful data such as the distances of crucial interactions they just identified in the solved structure, these distances are the really important ones since one shall expect to see the ligand recognition mechanism mainly around the head group rather than the universal acyl tail. Only RMSD plots of LysoPS and some snapshots of hydrogen bonds are not convincing. Besides these, if authors still want to present the RMSD data, they should provide the different RMSD profiles using the head group and the tail of LysoPS respectively to study the stability of binding pose.

6. The authors didn't show if the system itself is stable during their simulations, therefore I can't tell if the simulations results are reliable. Besides, 100ns seems too short for me as GROMACS can do pretty good job in term of performance given the size of the simulation box (100k atoms). I suggest the authors run their simulations longer (if there is no difficulty to do so), and provide some evidence showing the system is well equilibrated within the bilayer, then further analyses of stability of ligand binding will make sense.

7. The quality of Figure S5.a is very poor, it must be improved although it is not the main figures. The panels at least should have proper axes, ticks and labels. Besides, why authors picked those snapshots, it doesn't look like a random selection to me, and some of them I don't even know where they sit in the profiles. The authors suggested some water-mediating interactions, is there any noticeable density found in the cryo-EM map? It's not surprised to me that there are some water-mediating interaction networks, but a map with such good resolution must have some trace of water density and author should pay attention to this as well.

8. Using Alphafold model in docking is controversial, the authors didn't provide the quality of their Alphafold models to show how reliable they are, they should be included in the supplementary data. The results of rigid docking are also missing, how could one believe the presented poses are best poses without seeing data of energy function profiles of different poses? Also, what are the charged states for those titratable residues in their docking study, the method section should be re-written with more details, more suggestions are in my minor suggestions.

9. The author claimed "the N-acylation to the serine amine will interfere the polar network interactions" from the structural viewpoint to explain origin of antagonistic behavior. First, in absence of the acylated LysoPS binding structure, they should conduct the soft docking this time by allowing side chains of receptor to move to test if the interactions are interfered dramatically, and get the conclusion of how the acylation impacts the binding. Second, and more importantly, the loss of binding affinity of weakened binding is not equal to changing a ligand from agonist to antagonist, which should be more rationalized in the discussion.

Minor suggestions:

1. There are some spots according to the map in which water molecule should be placed, for example, density bubble between the O27 atom of lysoPS and carbonyl oxygen of L271, and the density bubble in the pocket surrounded by Q68, M102, S105, F243 and N284. One water molecule was predicted at the first site according to their initial MD group file, but the second wasn't which caused the collapse the the pocket in their model and that should be improved.

2. Sometimes, the figures and their captions are too simple. For example, the middle of Figure 1 is the molecular structure of LYSOPS (18:0) but wasn't revealed in the caption, besides, the LysoPS is too small. My suggestion is to put the LysoPS vertically and make the bonds more thicker and molecule larger. In Figure 3.b, those interactions label with yellow dashed lines shall be mentioned as effective h-bonds or salt-bridges in caption. Figure 3.c seems to be the main interaction network around the head group of LysoPS, the caption then needs to be rephrased as something like "An interaction plot of LysoPS head group and key residues of receptor ligand binding pocket". All interaction types defined by LigPlot need to be clarified. In Figure 3.d, the authors need to verify which statistical method/test they used to obtain the significance.

3. In the bottom table in Figure 3.b, the author labeled some residues in red and claimed they are conserved, but they may only be conserved among the certain group that bind LysoPS, otherwise the Histidine at the corresponding position of 2.64 is at same level of conservation, and so is the glycine at the corresponding position of 4.64.

4. The authors used "hydrogen bond" and "hydrogen-bond", it should be consistent. The authors should give more details of the simulation protocols, are they the standard ones from CHARMM-GUI? What water model they used? Why having so large system while they only simulated the GPCR (~300aa) in POPC bilayer?

5. I don't fully understand the statement of "titratable residues were protonated at their dominant states under neutral condition except for D65", for example, are all glutamate residues also protonated? The authors need to write it more clearly if those charged residues at normal pH are prepared with the right protonation states.

6. The authors need to provide the .mdp file for the MD simulation, topology file of LysoPS, and .pdb file for the system and deposit them into an accessible place to achieve the reproducibility.

7. Will the unique Gs engagement be possibly linked with the "NanoBiT tethering strategy" they adopted? It's worth discussing this topic in the discussion section.

REVIEWER COMMENTS

Reviewer #1 (Remarks to the Author):

This is a very interesting paper, and the authors have addressed all my concerns in the first review, I would thus support its publication.

Reviewer #2 (Remarks to the Author):

This is the revised version of the manuscript which I reviewed. The authors clearly addressed to my queries about the missing interaction of K98 with LysoPS, and a character of the interaction of TM6/ICL3 with Galpha protein. I believe theses additions of the discussions make this work much more scientifically rigorous. I agree with publication of this work in Nature Commutations without further change.

Reviewer #3 (Remarks to the Author):

In the paper “Structural basis of lysophosphatidylserine receptor GPR174 ligand recognition and activation” by Liang et al., one good resolved cryo-EM structure of GPR173/GS complex bound with LysoPS (18:0) was presented using method called NanoBiT tethering strategy. The overall resolution of $\sim 2.8\text{\AA}$ was good enough to capture those crucial interactions between GPCR and ligand (lysoPS) as well as with other G proteins. The unique ligand binding mode and spatial engagement of GS protein within the complex were discussed in this manuscript decently. Molecular dynamics (MD) simulations of GPCR/lysoPS complex in POPC bilayer were conducted based on the suggestions from the last round of peer-review. Overall, my opinion of this manuscript is that it's acceptable to be published in Nature Communications after some examinations of the data are taken carefully to support the arguments they proposed. Here are my major concerns and minor suggestions.

Thank you for your positive comments on our study.

Major concerns:

1. *The density map of the cryo-EM structure of Sphingosine 1-phosphate receptor 1 / Gi complex bound to S1P (PDB ID: 7WF7) is clearly not supporting the model the authors presented in Figure 3.c in terms of the orientation of side chains of all of those residues in the panel. The authors should refine model of 7WF7 based on the map to obtain more sensible binding pose and interactions otherwise the comparison and discussion are not correct.*

We accidently used an intermediate model of 7WF7, which our group had analyzed in a previous study. As stated in the S1PR1 paper, there is a dynamic in the S1P-bound S1PR1. We have now changed to 7VIE (PNAS 2022, doi:10.1073/pnas.2117716119), which has similar head interaction, and this will not change our conclusion that GPR174 has a unique ligand engaging mode (Fig. 1 of rebuttal letter and new Fig. 3c).

Fig. 1 of rebuttal letter. S1P phosphate head interaction in S1PR1.

2. In the section of discussing the activation mechanism of GPR174, the authors using a group of comparisons between the active form of GPR174 and an inactive form of P2Y1. However, how closely these two receptors are related? More importantly, comparing a cryo-EM structure solved using single-particle technique with a crystal structure with heavy crystal packing is tricky, one sample of related receptor with antagonist binding is also not enough. More structures shall be involved in this discussion.

We agree with the Reviewer #3 that a comparison of cryo-EM structure with crystal structure is not perfect. However, it is generally accepted that such comparison does generate useful information of receptor activation (Cell 2020, 10.1016/j.cell.2020.08.024; Nature 2020, 10.1038/s41586-020-2379-5; Nature 2020, 10.1038/s41586-020-2019-0; Nature 2019, 10.1038/s41586-018-0219-7) even when information of only active GPCR is available. A PDB search showed that the closest inactive receptor to GPR174 is P2Y1 and their amino-acid similarity in the conserved region is 51% according to the GPCRdb (<https://gpcrdb.org>). We agree with you that one sample of related receptor is not enough, for this purpose, we also compare active GPR174 with closely related inactive lipid receptors: ML056-bound S1PR1 (35% amino-acid similarity) and ONO-97808307-bound LPAR1 (32% amino-acid similarity). The comparison shows that the outward movement of TM6, the titling of Y292 (the NPxxY motif) to the center of the cavity and the movement of R116 (the DRY motif) toward the center of receptor as the signatures of receptor activation (Fig. 2 of rebuttal letter and new Fig. 4), consistent with previous analysis.

Fig. 2 of rebuttal letter. Activation of GPR174.

3. *Still about the inactivation form, since the authors have already noticed that the acylation of the L-serine of LysoPS will antagonize the receptor, had the authors tried to solve the structure of GPR174 with the acylated LysoPS?*

Solving the inactive structure is challenging because of the small size of the protein (GPCR alone as compared with active GPCR-G-protein structure). We unfortunately say that it is out of the scope of this study, but it will be an interesting project in future to further understand how small change in ligand behaves differently.

4. *The most of my concerns come from the MD part, the authors conducted MD simulations with three replicas to study if the binding mode of ligand in GPR174 is stable and energetic favorable. However, the authors didn't discuss this topic sufficiently, no energy calculation was conducted but three RMSD profiles of LysoPS. More importantly, several interactions observed in cryo-EM structure was lost in the initial model they used in MD simulation, for example, the interaction between Y79 and amine-group of L-serine in head group, as well as the interaction F169 and that amine group, which is due to the change of the position of L-serine in pocket. The authors should run proper equilibration before the production MD to at least maintain the binding pose first then discuss the stability of ligand.*

Thank you for your suggestion. As for energy calculation, we expect that it is beyond the scope of the study and realistic computational resources. Since LysoPS is a lipid, tail fluctuation will be a matter for the calculation and we need to calculate lipid solubility into both membrane and solution. Naive computations could be technically performed, but it will be less reliable and more sophisticated computations will be worth a paper (*e.g.*, [10.1021/acs.jcim.1c01147](https://doi.org/10.1021/acs.jcim.1c01147)).

For the ligand fluctuation, we re-examined the equilibration step (see revised method section for detail). We did observe system volume decreases and preservations in NPT-runs (Fig. 4 of rebuttal letter) as Miao and McCammon did ([10.1073/pnas.1614538113](https://doi.org/10.1073/pnas.1614538113)), so we believe that our simulations were performed after proper equilibration (see our response to #6 for quantitative discussion). Moreover, we speculate that the fluctuation of lipid ligands is the common nature in lateral access lipid receptors and that the binding position of LysoPS would not be fixed. Indeed, in a study of a closely related LPA6 receptor and its agonist LPA ([10.1371/journal.pone.0263296](https://doi.org/10.1371/journal.pone.0263296)), molecular dynamics simulations revealed LPA transitions between several states even though LPA6 receptor strongly recognizes the negatively charged phosphate group at the LPA head (Fig. S8 for comparison). In our MD simulation, we also observed the dynamic nature of the LysoPS molecule in the ligand pocket, which reflects the fluctuation of lipid ligands in general.

Fig. 4 of rebuttal letter. Box sizes before and after equilibration.

Solid lines show unit cells before (gray, $\sim 96 \times 96 \times 124 \text{ \AA}$) and after (colored, three independent runs, up to $94.1 \times 94.1 \times 118.4 \text{ \AA}$) equilibration.

5. *The tail of LysoPS was shown to be flexible which makes sense to me since it would partially interact with the bilayer lipids according to the position of LysoPS in Figure 1, however, the authors didn't report the meaningful data such as the distances of crucial interactions they just identified in the solved structure, these distances are the really important ones since one shall expect to see the ligand recognition mechanism mainly around the head group rather than the universal acyl tail. Only RMSD plots of LysoPS and some snapshots of hydrogen bonds are not convincing. Besides these, if authors still want to present the RMSD data, they should provide the different RMSD profiles using the head group and the tail of LysoPS respectively to study the stability of binding pose.*

We added distance plots of critical interactions that were confirmed by structure and mutant study (Y22, R75, K98, and R156) (Fig. 5 of rebuttal letter and as in Fig. S5d of the revised manuscript). In our three replicas of the MD simulations, R156 mostly retained contact with the LysoPS headgroup while the rest of the three residues exhibited both contact and non-contact distances. We also analyzed RMSD scoring and added various profiles of the head group, tail group, and overall structure (Fig. 6 of rebuttal letter and as in Fig. S5b of the revised manuscript). This analysis shows higher dynamics of the acyl-chain part than the head group.

Fig. 5 of rebuttal letter. Interactions between GPR174 and LysoPS. Distances between GPR174 and nearest heavy atoms in LysoPS were shown as traces. Same figures were added as Fig. S5d in the revised manuscript.

Fig. 6 of rebuttal letter. RMSD values of LysoPS. Traces indicate RMSD values for overall LysoPS (blue), tail acyl chain (orange), and non-acyl chain (green). Same figures were added as Fig. S5b in the revised manuscript.

6. The authors didn't show if the system itself is stable during their simulations, therefore I can't tell if the simulations results are reliable. Besides, 100ns seems too short for me as GROMACS can do pretty good job in term of performance given the size of the simulation box (100k atoms). I suggest the authors run their simulations longer (if there is no difficulty to do so), and provide some evidence showing the system is well equilibrated within the bilayer, then further analyses of stability of ligand binding will make sense.

We extended our simulations to 500 ns and added supplementary figures to show the receptor RMSD (Fig. 7 of rebuttal letter and as in Fig. S5a of the revised manuscript).

Fig. 7 of rebuttal letter. RMSD values during the simulations.

RMSD values of C α atoms in GPR174 were calculated and plotted. Same figures were added as Fig. S5a-c in the revised manuscript.

We validated the equilibration step by the following criteria: (i) Total energy (Fig. 8a of rebuttal letter), (ii) Temperature (Fig. 8b of rebuttal letter), (iii) Pressure (Fig. 8c of rebuttal letter). All of these values converged at the specified values, suggesting the system had been well equilibrated.

Fig. 8 of rebuttal letter. Total energy, temperature, and pressure during the equilibration steps.

Values during the equilibration steps were plotted on gray background for NVT equilibration and on white background for NPT equilibration. Horizontal gray lines indicate specified values for temperature (310 K, **b**) and pressure (1 bar, **c**). Note that the traces of the runs 1-3 overlapped and thus are not visible in **a** and **b**.

7. The quality of Figure S5.a is very poor, it must be improved although it is not the main figures. The panels at least should have proper axes, ticks and labels. Besides, why authors picked those snapshots, it doesn't look like a random selection to me, and some of them I don't even know where they sit in the profiles. The authors suggested some water-mediating interactions, is there any noticeable density found in the cryo-EM map? It's not surprised to me that there are some water-mediating interaction networks, but a map with such good resolution must have some trace of water density and author should pay attention to this as well.

We apologize for the broken figures that appeared during file handling. Based on the Reviewer #3's comment, we replaced the figure with the new one as below (Fig. S5b in the revised manuscript).

As for the snapshots, we chose them at manually selected timings where RMSD of the LysoPS molecule seemed to be stable (e.g., 40.0 ns, 50.0 ns, and 80.0 ns) in the previous manuscript. Yet, we agree with the Reviewer #3 that we rely on a rational method to pick snapshots. We clustered trajectories by the relative position of non-acyl chain moiety in LysoPS to select centroidal snapshots (Fig. 9 of rebuttal letter and as in Fig. S5c in the revised manuscript).

Fig. 9 of rebuttal letter. Representative snapshots in simulations.

Trajectories were clustered by the relative position of non-acyl chain moiety in LysoPS. Inset figures show centroidal snapshots. Same figures were added as Fig. S5c in the revised manuscript.

Regarding the water density, we carefully examined the map of GPR174. Indeed, we found a water density on the upper ligand binding pocket (Fig. 10 of rebutter letter and updated Fig. 2b), and this water molecule connects LysoPS with R18 via hydrogen bond network interaction, suggesting that water molecules do play roles in the ligand receptor interaction.

Fig. 10 of rebuttal letter. A water molecule connects LysoPS and R18.

8. Using AlphaFold model in docking is controversial, the authors didn't provide the quality of their AlphaFold models to show how reliable they are, they should be included in the supplementary data. The results of rigid docking are also missing, how could one believe the presented poses are best poses without seeing data of energy function profiles of different poses? Also, what are the charged states for those titratable residues in their docking study, the method section should be re-written with more details, more suggestions are in my minor suggestions.

We are sorry for the confusion. Actually, we used a combination of AlphaFold, Rosettafold (Science 2021, 10.1126/science.abj8754) and the active GPR174 for modeling GPR34 and P2Y10 as the AlphaFold predictions of most GPCRs are in inactive conformation which is not optimal for agonist docking. In the this case, alphafold of GPR34 (AF-Q9UPC5) and rosettafold of active GPR34 (Q9UPC5_model.active), as well as alphasfold of P2Y10 (AF-O00398) and rosettafold of active P2Y10 (O00398_model.active), and the active GPR174 were used as models for the active GPR34 and P2Y10 modeling via modeller10.0. (we added detail in revision). Then the ligand was docked into the active receptor, we also include all top 5 poses of the docking, as well as the statistics (Fig. 11 of rebuttal letter), we noticed that P2Y10 show a better docking outcome than GPR34, this may be due that P2Y10 is much closer to GPR174 than GPR34 (74% vs 43%), so the model of P2Y10 is more accurate than GPR34. The charged states of the pocket residues were determined by Protonate3D (Proteins 2009, 10.1002/prot.22234) in the MOE package.

Fig. 11 of rebuttal letter. Docking of LPS into GPR34 and P2Y10.

9. The author claimed “the N-acylation to the serine amine will interfere the polar network interactions” from the structural viewpoint to explain origin of antagonistic behavior. First, in absence of the acylated LysoPS binding structure, they should conduct the soft docking this time by allowing side chains of receptor to move to test if the interactions are interfered dramatically, and get the conclusion of how the acylation impacts the binding. Second, and more importantly, the loss of binding affinity of weakened binding is not equal to changing a ligand from agonist to antagonist, which should be more rationalized in the discussion.

Thank you for your suggestion. We have docked cpd8a which has an acylation on the serine amine to GPR174 by the induced-fit-in model, the docking shows that cdp11 adopted a similar binding mode of LPS. However, we observed a clear loss of polar interaction with Y79, R75 and F169 in cpd8a (Fig. 12 of rebuttal letter, left panel, white circled area, also Supplementary of Fig. 6d). In the meantime, we also observed a gain of polar interaction of Y22 and H274 (Fig. 12 of rebuttal letter, left panel, green circled area, also Supplementary of Fig. 6d). Thus, there is a switch of binding, not a loss of binding. We note that we did not claim that weakened binding equal a change

from agonist to antagonist in our paper. Regarding how the switch of binding mode change conformation of receptor from active to inactive, this cannot be simply addressed by docking, and can only be addressed by an antagonist-bound receptor structure, which will be a separated story.

Fig. 12 of rebuttal letter. Docking of an acylated N-serine amine derivative of LysoPS into GPR174.

Minor suggestions:

1. There are some spots according to the map in which water molecule should be placed, for example, density bubble between the O27 atom of lysoPS and carbonyl oxygen of L271, and the density bubble in the pocket surrounded by Q68, M102, S105, F243 and N284. One water molecule was predicted at the first site according to their initial MD group file, but the second wasn't which caused the collapse the the pocket in their model and that should be improved.

Thank you very much for your careful examination of our map. Yes, you are right, after examination of the map, we do find the two water molecules densities. The first water molecule has been discussed in response to your major point 7 (Fig. 10 of rebuttal letter). The second water molecule makes hydrogen bond interaction with Q68 (Fig. 13 of rebuttal letter). We have added the water molecule into our model.

Fig. 13 of rebuttal letter. A water molecule interacts with Q68.

2. Sometimes, the figures and their captions are too simple. For example, the middle of Figure 1 is the molecular structure of LYSOPS (18:0) but wasn't revealed in the caption, besides, the LysoPS is too small. My suggestion is to put the LysoPS vertically and make the bonds more thicker and molecule larger. In Figure 3.b, those interactions label with yellow dashed lines shall be mentioned as effective h-bonds or salt-bridges in caption. Figure 3.c seems to be the main interaction network around the head group of LysoPS, the caption then needs to be rephrased as something like "An interaction plot of LysoPS head group and key residues of receptor ligand binding pocket". All interaction types defined by LigPlot need to be clarified. In Figure 3.d, the authors need to verify which statistical method/test they used to obtain the significance.

Thank you for your suggestion, we have made the suggested changes on Fig. 1, Fig. 2c (we assume that Fig.2c, 2d were erroneously written as "Figure 3.c, 3d"). We included the statement of the statistical analysis as "statistical analyses were performed using the ordinary one-way ANOVA followed by the two-sided Sidak's post hoc test with the expression-matched (colored) WT response. ns, $p > 0.05$; ** $p < 0.01$; *** $p < 0.001$." (Figure 2 legend).

3. In the bottom table in Figure 3.b, the author labeled some residues in red and claimed they are conserved, but they may only be conserved among the certain group that bind LysoPS, otherwise the Histidine at the corresponding position of 2.64 is at same level of conservation, and so is the glycine at the corresponding position of 4.64.

Thank you for your correction. We have modified the description of Fig. 3b to show that the red color marked conserved residues of LysoPS-binding receptors.

4. The authors used "hydrogen bond" and "hydrogen-bond", it should be consistent. The authors should give more details of the simulation protocols, are they the standard ones from CHARMM-GUI? What water model they used? Why having so large system while they only simulated the GPCR (~300aa) in POPC bilayer?

Thank you for your suggestion, we have unified to "hydrogen bond". Briefly, our simulations followed the default protocols from CHARMM-GUI, and we also added more details in the revised manuscript. Since we did not plan to run long simulations for GPR174, we did not particularly care about the size as long as GROMACS performs well.

5. I don't fully understand the statement of "titratable residues were protonated at their dominant states under neutral condition except for D65", for example, are all glutamate residues also protonated? The authors need to write it more clearly if those charged residues at normal pH are prepared with the right protonation states.

We apologize for the inaccurate statements. We believe the revised method details reflect what we really intended. Key protonation processing “titratable residues were kept at their dominant states under neutral conditions, except for D652.50”. For details, see the method (line 474-484)

6. The authors need to provide the .mdp file for the MD simulation, topology file of LysoPS, and .pdb file for the system and deposit them into an accessible place to achieve the reproducibility.

In addition to the revised method details, we have prepared these files to Zenodo (10.5281/zenodo.7364300). These files will be public after the release of the experimental GPR174 structure.

7. Will the unique G_s engagement be possibly linked with the “NanoBiT tethering strategy” they adopted? It’s worth discussing this topic in the discussion section.

We do not think that the unique G_s engagement of GPR174 is caused by the NanoBiT strategy. The NanoBit strategy can facilitate or stabilize the weak interaction by increasing local concentration of interaction partners, but cannot change the interaction nature, and cannot force an interaction that does not exist in nature. A comparison of our recently GPR110/G_s complex (Nature Communications 2022, 10.1038/s41467-022-33173-4) with the GPR110/G_s complex without NanoBit strategy (Nature 2022, 10.1038/s41586-022-04580-w) shows that the G_s engagement is same, the Nanobit strategy only increases the global resolution of the complex.

Reviewers' Comments:

Reviewer #3:

Remarks to the Author:

The authors have answered most of my concerns and questions in this revision, the new analyses have been done will make this work more solid I believe. I thank the authors for their great responses.

But I still have two questions about the data they present in Figure 5 and 6 of rebuttal letter, in the main text of manuscript (Line 151-152), the author claimed "Importantly, the key residues identified by the mutant study (R75 2.60, K98 3.32 and R156 4.64) are all involved in ligand recognition in the MD simulation.", but according to the data in Fig.5 of rebuttal letter, only R156 forms stable interactions with the LysoPS, maybe K98 as well (I not very sure due to the fluctuation), the other two residues Y22 and R75 are almost not interacting with LysoPS in all three replicas after 250ns. I am not sure what the authors want to show in terms of supporting "ligand recognition" was observed in MD, such unstable interactions seen in MD are certainly not supporting any ligand recognition, especially the interactions in the cryo-em structure are so clear, the flexibility of LysoPS is not the reason/excuse otherwise the other kind of head group in other Lyso-lipid will also bind. Any proposed interactions/residues play vital roles in ligand recognition must be stable and unique to certain head group, especially the authors had claimed the binding mode is so unique.

In Fig.6 of rebuttal letter, the authors presented the RMSD of LysoPS, acyl tail of LysoPS and non-acyl tail in LysoPS against time for three replicas, and they said "This analysis shows higher dynamics of the acyl-chain part than the head group.", however, except the replica 1 in which the non-acyl part has lower RMSD value than acyl part, the acyl part is having lower RMSD value comparing with the non-acyl part in both replica 2 and 3 after 300ns, which is showing that the head group is less stable than the acyl-chain. The authors need to explain this and the last point I mentioned.

REVIEWER COMMENTS

Reviewer #3 (Remarks to the Author):

The authors have answered most of my concerns and questions in this revision, the new analyses have been done will make this work more solid I believe. I thank the authors for their great responses.

Thank you for your positive comments for our revision.

But I still have two questions about the data they present in Figure 5 and 6 of rebuttal letter, in the main text of manuscript (Line 151-152), the author claimed “Importantly, the key residues identified by the mutant study (R75 2.60, K98 3.32 and R156 4.64) are all involved in ligand recognition in the MD simulation.”, but according to the data in Fig.5 of rebuttal letter, only R156 forms stable interactions with the LysoPS, maybe K98 as well (I not very sure due to the fluctuation), the other two residues Y22 and R75 are almost not interacting with LysoPS in all three replicas after 250ns. I am not sure what the authors want to show in terms of supporting “ligand recognition” was observed in MD, such unstable interactions seen in MD are certainly not supporting any ligand recognition, especially the interactions in the cryo-em structure are so clear, the flexibility of LysoPS is not the reason/excuse otherwise the other kind of head group in other Lyso-lipid will also bind. Any proposed interactions/residues play vital roles in ligand recognition must be stable and unique to certain head group, especially the authors had claimed the binding mode is so unique.

By following the Reviewer #3’s suggestion, we have changed the text of the manuscript to focus on R156 as below. In the previous revised manuscript, we originally intended to describe the van der Waals interactions with LysoPS to form the cavity. However, as pointed out by the Reviewer #3, these interactions included weak, transient ones. We thus toned down for relatively unstable interactions and described stable interactions as below.

Original text: “*Importantly, the key residues identified by the mutant study (R75^{2.60}, K98^{3.32} and R156^{4.64}) are all involved in ligand recognition in the MD simulation.*”

Updated text: *“Especially, the key R156^{4.64} residue retained the direct hydrogen bond with LysoPS as implied by the experimental structure and mutant study. The other key residues identified by the mutant study (Y22^{1.35}, R75^{2.60} and K98^{3.32}) showed weaker interactions than R156^{4.64}, suggesting their other roles in complicated ligand recognition.”*

In Fig.6 of rebuttal letter, the authors presented the RMSD of LysoPS, acyl tail of LysoPS and non-acyl tail in LysoPS against time for three replicas, and they said “This analysis shows higher dynamics of the acyl-chain part than the head group.“, however, except the replica 1 in which the non-acyl part has lower RMSD value than acyl part, the acyl part is having lower RMSD value comparing with the non-acyl part in both replica 2 and 3 after 300ns, which is showing that the head group is less stable than the acyl-chain. The authors need to explain this and the last point I mentioned.

We apologize for the incorrect description. During our revision we initially intended to show the intramolecular fluctuation of LysoPS. However, we omitted such figure during the process and did not include it in the rebuttal letter. In Fig. 6 of the previous rebuttal letter (or Fig. S5 in the revised manuscript), we aligned with the receptor C α atoms and then calculated the ligand RMSD. Since the procedure does not directly evaluate the intramolecular fluctuation, we added the notion to clarify the head group drift in simulations as below.

Original text: *“The polar head group tended to form hydrogen bonds with or without water molecules from the extracellular side.”*

Updated text: *“The relative position of the non-acyl chain moiety, including the polar head group, tended to drift within the positively charged cavity.”*